**Data Availability Statement:** All relevant data are within the paper and its Supporting Information files.

# *Ex vivo* investigation on internal tunnel approach/internal resin infiltration and external nanosilver-modified resin infiltration of proximal caries exceeding into dentin

**Andrej M. Kielbassa** [1]*, **Marlene R. Leimer**[1], **Jens Hartmann**[2], **Stephan Harm**[2], **Markus Pasztorek**[2], **Ina B. Ulrich**[1]

**1** Centre for Operative Dentistry, Periodontology, and Endodontology, University of Dental Medicine and Oral Health, Danube Private University, Krems, Austria, **2** Department for Biomedical Research, Danube University, Krems, Austria

* andrej.kielbassa@dp-uni.ac.at

## Abstract

This *ex vivo* proof-of-concept study aimed to investigate the effect of nanosilver particles (AgNP) added to a conventional infiltrant resin (Icon) on external penetration into natural proximal enamel caries exceeding into dentin after internal tunnel preparation and internal infiltration. Carious lesions (ICDAS codes 2/3) of extracted human (pre-)molars revealing proximal caries radiographically exceeding into dentin (E2/D1 lesions) were preselected. Then, 48 of those specimens showing demineralized areas transcending the enamel-dentin border as assessed by means of near-infrared light transillumination (DIAGNOcam) were deproteinized (NaOCl, 5%). Using an internal tunnel approach, occlusal cavities central to the marginal ridge were prepared. Excavation of carious dentin, total etch procedure ($H_3PO_4$, 40%), and internal resin infiltration (FITC-labeled) followed, along with final restorations (flowable composite resin). Outer lesion surfaces were etched (HCl, 15%) prior to external infiltration (RITC-labeled). Group 1 (control; n = 24) used non-modified infiltrant, while an infiltrant/AgNP mixture (20 nm; 5.5 wt%) was used with experimental Group 2 (n = 24). Non-infiltrated pores of cut lesions were stained (Berberine), and specimens were analyzed using confocal laser scanning microscopy. Compared to the non-filled infiltrant, incorporation of AgNP had no effect on the resin's external penetration. Between the groups, no significant differences regarding internal or external infiltration could be detected, and non-infiltrated lesion areas did not differ significantly (p>0.109; *t*-test). The internal tunnel preparation in combination with both an internal resin infiltration and an additional external infiltration approach using a nanosilver-modified infiltrant resin leads to increased infiltrated lesion areas, thus occluding and adhesively stabilizing the porous volume of the demineralized enamel. While exerting antimicrobial effects by the nanosilver particles, this approach should have the potential as a viable treatment alternative for proximal lesions extending into dentin, thus avoiding the sacrifice of sound enamel, postponing the frequently inevitable restoration/re-restoration cycle of conventional proximal caries treatment, and improving dental health.

**Funding:** Regardless of the support from the authors and their respective institutions, no specific external funding was available for this investigator-driven ex vivo study.

**Competing interests:** Andrej M. Kielbassa is appointed as inventor in Austrian, Brazilian, Canadian, Chinese, French, German, Indian, Italian, Japanese, Korean, Russian, Swiss, UK and US patents (held by Charité - Universitätsmedizin Berlin, Germany) for the infiltration technique for carious lesions ("Method and means for infiltrating enamel lesions", Patent Number: 8853297); these patents have been licensed by DMG (Hamburg, Germany), and Andrej M. Kielbassa receives royalties from this license. This does not alter our adherence to PLOS ONE policies on sharing data and materials.

## Introduction

Minimal intervention is an indispensable element of modern dentistry focusing on preventive or non-surgical actions to preserve dental hard substances, thus avoiding any unnecessary sacrifice of tooth tissues, and ensuring a longest possible tooth survival [1]. Introduced in 2009, the resin infiltration technique using a low-viscosity resin originally has been developed with the intention to penetrate the demineralized and porous inter-crystalline spaces of initial subsurface enamel lesions, thereby occluding the latter after polymerization [2]. This ultraconservative approach effectively builds a covalently bound three-dimensional polymer framework [3, 4], thus (partially) replacing the lost minerals, encapsulating the hydroxyapatite crystals, micromechanically interlocking the remaining enamel prisms, and acting as an effective barrier for hydrogen ions to inhibit further demineralization and to arrest proximal subsurface lesion progress [4, 5].

Accordingly, timely systematic reviews on the clinical efficacy of this micro-invasive solution have confirmed its efficacy [6, 7]. With observation periods of up to four years in a clinical trial [8], the resin infiltration of proximal enamel lesions would seem to complement (or even outperform) other interventions like fluoridation and improved interdental hygiene [6, 9], even with high-risk caries patients [10, 11]. In fact, this approach does prevent from overtreatment, thus underpinning the suggested concept of a drill-less approach [12]. Consequently, a recently published guideline summarizing the respective literature has concluded that the resin infiltration concept is clinically feasible and reliable, and offers high success rates with non-cavitated proximal caries lesions restricted to enamel [13]. By implementing this technique, dental health will be maintained, and the use of surgical intervention will be reduced to a minimum, thus following the recommendations adopted by the International Caries Classification and Management System (ICCMS™) [14] and by CariesCare International (CCI™) [15].

Notwithstanding, initial enamel lesions are characterized by a reduced mineral content, thus leading to a decreased microhardness [16], and this in turn will result in a decreased stability, possibly jeopardizing the integrity of the respective regions; resin infiltration, however, has been reported to increase surface microhardness of demineralized human [17–19] or bovine [20–22] enamel significantly. All in all, this would suggest a recovered surface resistance of the respective areas [23]; nevertheless, it should be kept in mind that in particular with progressed enamel carious lesions the infiltration frequently will be inhomogeneous [24] and incomplete [3, 24] with respect to the total lesion depth. Not astonishingly, one of the recently published studies focusing on this topic could not reveal a reestablished microhardness considered comparable to sound enamel [25]. Thus, the infiltrated surface has not been shown to completely resist new cariogenic challenges [25, 26], and a previous review has emphasized that for caries extending into dentin, treatment efficacy of resin infiltration was not significantly different from the non-infiltrated controls [7]. Additionally (and not unexpectedly), a recently published randomized clinical trial has confirmed that the resin infiltrant's capacity to arrest caries progression of lesions reaching the outer dentin is reduced to 64% [10], thus suggesting an only poor efficacy of resin infiltration for these advanced lesion types.

Obviously, surface microhardness is an inadequate single parameter to conclusively assess infiltrated caries lesions, and cross-sectional microhardness evaluations might be more meaningful [27]. Indeed, continuously decreasing cross-sectional hardness values of demineralized and infiltrated enamel have previously been shown with increasing lesion depths [23], and this would render non-infiltrated enamel lesion areas vulnerable. Moreover, it has recently been shown that the level of demineralization correlates with the presence of superficial microcracks within the vicinities of proximal contact areas [28]; hence, it does not seem surprising that the prevalence of marginal ridge fractures is associated with the presence of proximal carious

lesions [29]. To overcome this fragility, a more comprehensive attempt to adhesively reinforce the enamel lesion would seem favorable [30, 31], and, therefore, a combined external/internal infiltration concept increasing the amount of infiltrated lesion volume has recently been introduced [32]. This treatment approach should further strengthen lesion resistance to (micro-) fractures.

Reflecting on proximal caries lesions extending (radiographically) into dentin and intended to be treated should raise a further consideration, namely that of the macroscopically non-cavitated (but nevertheless pre-damaged) surface of proximal caries. Indeed, previous papers have clarified that both the tendency of surface breakdown [33, 34] and the ICDAS codes [35, 36] will increase with advanced radiographic lesion extensions. Due to the material's inherent mechanical properties [37], the infiltration approach using an unfilled and low-viscosity resin will not be able to completely fill up any (micro-)cavitations [2, 3, 38], nor will it be able to adequately smoothen the infiltrated surface of demineralized (and rough) enamel [39]; thus, several investigations have clearly elucidated that roughness of infiltrated lesions will remain increased if compared to sound enamel [18, 20, 21, 25, 40], and this will not be perfectible by various polishing procedures [41]. Rough surfaces (with $R_a$ values exceeding 0.2 μm as the critical threshold), however, are susceptible to facilitate biofilm accumulation, and this has been revealed for infiltrated surfaces as well, even if to a lesser extent if compared to non-infiltrated lesions [17, 19, 42].

Therefore, adding filler particles exerting antibacterial properties to the low-viscosity resin might be a promising enhancement of the infiltration approach, and this should be interesting in particular for deeper lesions (reaching radiographically beyond the enamel-dentin junction), thus accidentally or intentionally surpassing the originally recommended indications for resin infiltration of non-cavitated enamel caries scored as ICDAS 1 and 2 (International Caries Detection and Assessment System) [35]. One possible prospective filler candidate would seem silver nanoparticles (AgNP) [43], ranging from 1 to 100 nm in diameter; in recent years, AgNP have been increasingly used for a wide range of applications in (nano-)medicine, and successful dental implementations have been reported as well [44, 45], including composite resins [46, 47]. The desired antimicrobial effects have been revealed [48, 49], and especially the long-lasting [50] and the long-distance bactericidal capability [51] should qualify AgNP as complimentary additives to infiltrant resins, thus preventing the latter from microbial re-colonization.

However, the available literature does not provide any information on possible effects of incorporated AgNP on the penetration ability of a resin infiltrant (Icon Caries Infiltrant; DMG, Hamburg, Germany). Thus, our objective with the present *ex vivo* investigation was to modify the resin infiltrant by using AgNP, and we hypothesized ($H_0$) that the addition of AgNP would have no influence on the resin's external penetration and the overall (internal/external) infiltration ability into non- and micro-cavitated proximal caries lesions (exceeding the enamel-dentin junction) of human premolars and molars. This null hypothesis was tested against the alternative hypothesis ($H_1$) of a difference.

## Materials and methods

### Visual and radiographic selection of teeth

Extracted teeth were obtained from a company responsible for disposal of dental materials (Enretec, Velten, Germany), and human premolars and permanent molars showing chalky white/brownish incipient carious lesions on at least one proximal tooth surface were selected for the present study. Teeth revealing occlusal and/or proximal restorations as well as those showing visible fractures were excluded. In accordance with the German regulations of the Central Ethical Committee regarding the use of human body material in medicine [52], no

ethical approval was mandatory, and we had unrestricted permission for the use of these anonymous teeth in research and for publication.

Prior to further use, all specimens were carefully cleaned from soft tissues and calculus (and with special care to avoid touching the carious lesion) using a dental ultrasonic device (Teneo; Dentsply Sirona, Bensheim, Germany). All teeth were preselected (MRL), independently classified visually according to the International Caries Detection and Assessment System (ICDAS) [35], and consented by three experienced observers (AMK, IBU, MRL) under ideal lighting conditions (Sirona C8+; Dentsply Sirona) with the naked eye; careful and pressureless probing [53] using a dental explorer (EXD3CH6; Hu-Friedy, Chicago, IL, USA) was used to exclude teeth obviously revealing frank or deep cavitations. For the current investigation, only proximal surfaces revealing ICDAS codes 2 and 3 were selected, and all teeth were dried with paper towels. Subsequently, the proximal caries lesions were photographed (E-M5 Mark II; lens 60 mm, 1:2.8, ringflash STF-8; Olympus, Hamburg, Germany).

To complement the (simulated clinical) evaluation, all selected teeth (initially consisting of 182 premolars and 122 molars) were radiographed by means of a commonly used X-ray system (0.08 s for premolars/0.10 s for molars; 60 kV; 7 mA; Heliodent plus; Dentsply Sirona) [32, 39]. To guarantee a reproduction of the respective positions, a wooden holder (in-house production) was used in combination with a surface bed made out of silicone (Silaplast; Detax, Ettlingen, Germany), to fix the X-ray tube. All teeth were perpendicularly attached in a flexible silicone base (Silaplast; Detax), with a 4-cm distance between their buccal aspects and the X-ray tube [33]. To simulate soft tissues, three Perspex panels (with a total dimension of 15 mm; Perspex Distribution, Chelmsford, UK) were fixed between tube and tooth as described previously [33, 39]. Subsequently, the radiological lesion depths were examined and consented by three examiners (AMK, IBU, MRL) using a six-point classification system [54]; only teeth revealing either E2 (translucency in the inner half of enamel) or D1 lesions (translucency in the outer third of dentin) were chosen. A flowchart presenting the study set-up is given with Fig 1.

## Confirmation of dentin caries using near-infrared transillumination

In addition, all teeth were screened by using a digital imaging near-infrared light transillumination (NILT) device for caries detection (DIAGNOcam; KaVo, Vienna, Austria), to double check the radiographic outcome as well as to determine the extension of lesion depths, viewed from the intact occlusal surface [55]. The teeth were placed in a gingival mask (elastic replacement gingiva, AN-4 WUKV; Frasaco, Tettnang, Germany), thus imitating clinical conditions. To fade out near-infrared radiation and ambient light, black colored (Edding 3000, 3 mm black; Edding, Ahrensburg, Germany) artificial teeth (n = 2; Frasaco) were fixed both mesially and distally using a glue gun (PSM Bestpoint, Wels, Austria). Before using the DIAGNOcam (KaVo), all teeth were immersed into saline solution (in-house production) for mimicking of natural saliva. After this confirmatory examination, only teeth showing an indisputable translucency in the outer third of dentin (comparable to radiographically visible D1 lesions) as diagnosed by means of the NILT device were chosen [56], and a total of 48 teeth (24 premolars, 24 molars) were selected. Then, all teeth were randomly divided into 2 groups, each comprising the same number of molars and premolars as well as the same number of ICDAS code 2 (n = 12) and code 3 (n = 12) lesions (Fig 1). Until further usage, the teeth were stored in 0.9% sodium chloride solution (0.9% NaCl solution; in-house production) at room temperature.

## Preparatory steps

Subsequently, for all teeth the highest proximal demineralization areas of each tooth were defined with a calibrated laser fluorescence device (DIAGNOdent pen; KaVo) [32]. These

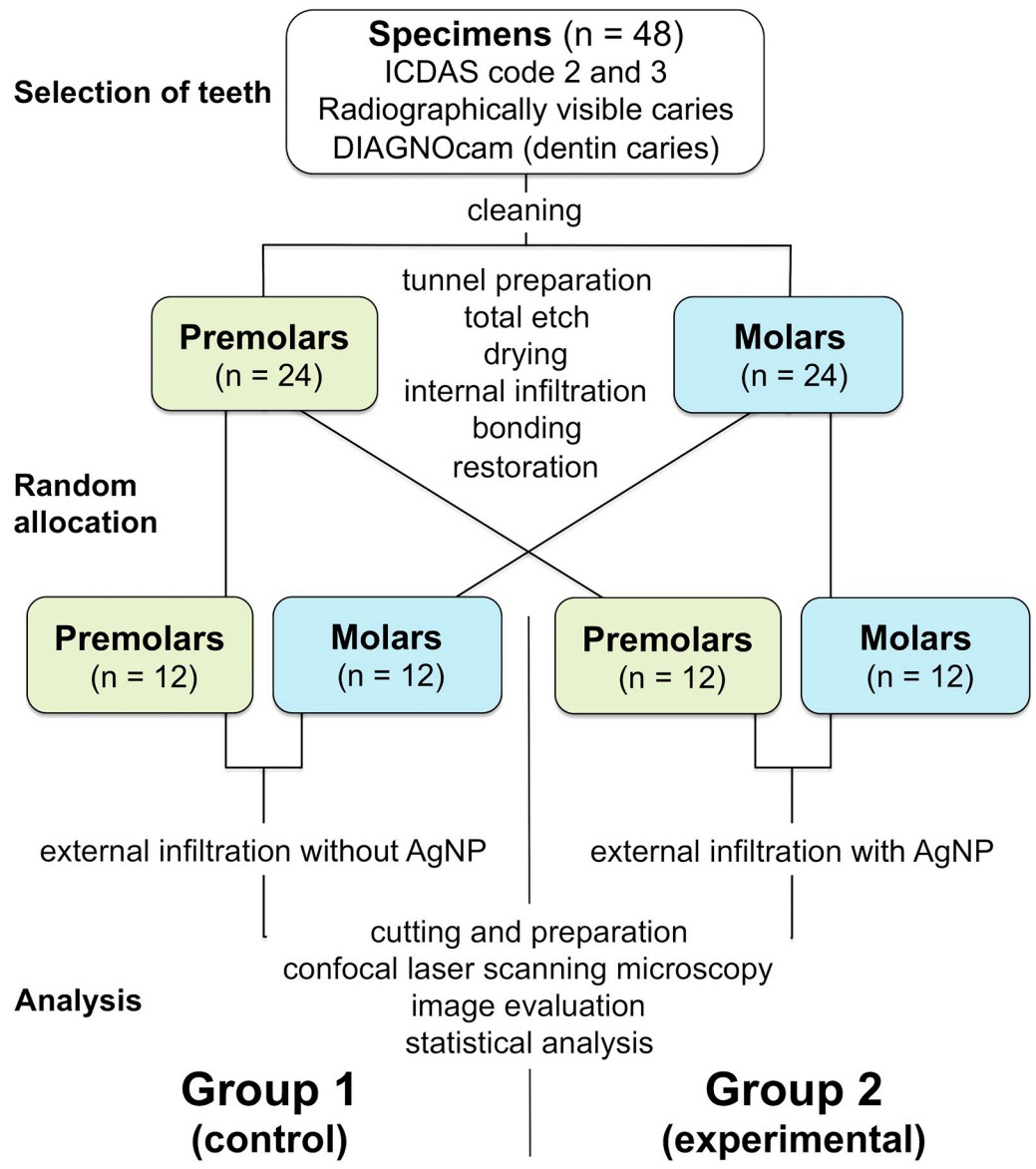

**Fig 1. Flowchart presenting group assignment and experimental set-up.**

areas were highlighted on printed photos of each lesion, and the respective marking points served as orientation for the cutting procedures described below. Finally, all chosen teeth were numbered and stored in hermetically sealed boxes (TO 706–12; Sogenex, Tood, Malo, Italy), filled with saline (in-house production) at room temperature. Fig 2 depicts a representative example of the clinical appearance (Fig 2A) of a specimen's proximal caries, along with the respective radiographic (Fig 2B) and NILT views (Fig 2C).

After the finalization of the described selection process, all teeth were put into a plastic cup (Drinking Cups, #900–8366; Henry Schein, Melville, NY, USA), which was previously filled up with sodium hypochlorite (5%, NaOCl solution; Apotheke zum goldenen Engel, Graz, Austria) for deproteinization (20 min) of the outer surfaces. Then, all teeth (n = 48) were thoroughly cleaned with water spray using a multifunctional syringe (Sprayvit, Teneo; Dentsply Sirona) for 30 s, to remove any residuals of hypochlorite and dissolved organic material.

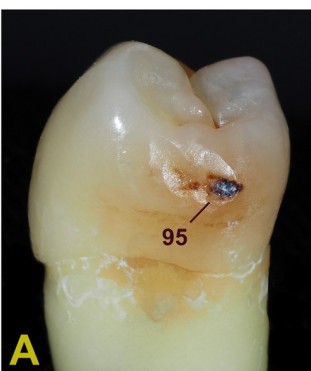 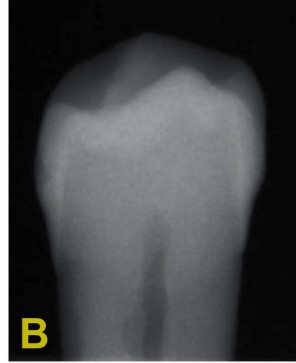 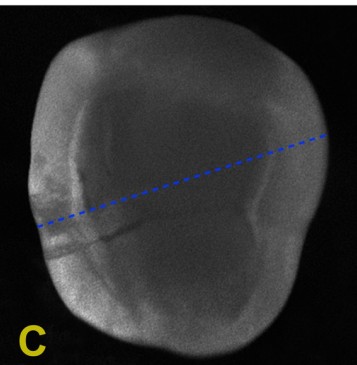

**Fig 2. Representative example of premolar revealing proximal caries.** Representative specimen of the experimental Group 2 (internal tunnel preparation as well as internal infiltration, and external infiltration using an infiltrant/ nanosilver particle mixture). (**A**) Macroscopic view of incipient proximal caries lesion before treatment (DIAGNOdent pen value measured as indicated). (**B**) Radiograph of the respective specimen, revealing the proximal caries lesion not clearly extending into dentin. (**C**) Corresponding radiation-free DIAGNOcam image of the same specimen, depicting the extent of the carious lesion, along with the sectional plane for CLSM evaluation (indicated by the blue dotted line).

## Occlusal access preparation and internal infiltration

Using a red contra angle handpiece (160,000 rpm, C200 L, 1:5; Dentsply Sirona) in combination with permanent water cooling, occlusal cavities central to the marginal ridge were prepared using a minimally invasive diamond-coated bur (#830.314.010; Komet Austria, Salzburg, Austria) until the proximal caries of all teeth became apparent from the internal side. Then, after removing the dentin layers, the inner portion of the proximal lesion was uncovered from the central cavity. To gain access to the diseased enamel, a rose head bur (#H1SE.204.016; Komet Austria) was used without water cooling, driven by a green handpiece (10,00 rpm, S-Max, M15L, 4:1; Dentsply Sirona). All procedures were carried out using magnifying glasses (opt-on TTL 2.7×, 400 mm working distance; orangedental, Biberach, Germany).

After this internal tunnel approach, the inner enamel surfaces (located at the depth of the occlusal cavity) of all teeth dried by means of a compressed air stream (Sprayvit, Teneo; Dentsply Sirona), followed by total etch using 40% phosphoric acid gel (HS Etch Gel; Henry Schein). The etchant was thoroughly removed using an air/water sprayer (Sprayvit, Teneo; Dentsply Sirona; 30 s) after an exposure time of 1 min. Subsequently, the cavity was fully dried with oil-free, compressed air (Sprayvit, Teneo; Dentsply Sirona; 30 s).

Prior to infiltration, the infiltrant (Icon Caries Infiltrant; DMG; 2 drops) was labeled using a green fluorescent dye (0.1 mmol fluorescein isothiocyanate, FITC; Babenberger Apotheke, Vienna, Austria). For exact dispensing, 10 µl of FITC were pipetted (Research plus–Physio-Care Concept; Eppendorf, Hamburg, Germany; Pipette tips epT.I.P.S, 200 µl; Eppendorf) into mixing pads (#9008146, HS-Mixing palette; Henry Schein). After 20 min, the alcohol was fully evaporated; then, the dye was mixed with 2 drops of the resin infiltrant using a micro-brush (Microbrush Plus, superfine white, Ø 1 mm; Microbrush International, Grafton, WI, USA). Subsequently, the FITC-labeled infiltrant was carefully applied onto the demineralized inner enamel for 3 min, in each case using a new micro-brush. The resin infiltrant was light-cured (by assuring a 3-mm distance; Mini LED Curing Light, >1,250 mW/cm$^2$; Satelec Acteon, Mérignac, France) via the prepared cavity for 40 s, and the resin infiltration procedure was repeated once (1 min), followed by a polymerization for another 40 s. Finally, the entire cavity was filled up with flowable, light-curing composite resin (G-Premio Bond; GC Europe,

Leuven, Belgium; and G-ænial Flo X shade *A3*; GC Europe), which finally was polymerized for 40 s (see flowchart, Fig 1).

## External etching and drying process

All outer surfaces of the proximal lesions were etched with hydrochloric acid gel (15% HCl, Icon-Etch; DMG; 2 min). Subsequently, the etchant was completely removed using an air/water sprayer (Sprayvit, Teneo; Dentsply Sirona) for 30 s; the teeth were then air-dried for another 30 s (Sprayvit, Teneo; Dentsply Sirona). Afterwards, a complete draining by means of ethanol (99%, Icon-Dry; DMG; 30 s) and oil-free, compressed air (30 s) followed.

## External infiltration of Group 1

All teeth of Group 1 (12 premolars/12 molars; n = 24; ICDAS code 2: n = 12, ICDAS code 3: n = 12) were externally infiltrated (Icon Caries Infiltrant; DMG) according to the recommendations as given by the manufacturer. The resin infiltrant for Group 1 was labeled with a red fluorescent dye (0.1 mmol rhodamine B isothiocyanate, RITC; Babenberger Apotheke). After infiltrating the proximal lesions for 3 min, surpluses were removed by means of dental floss (Oral-B Superfloss; Procter & Gamble, Schwalbach, Germany) without using the spongy floss part, and foam pellets (#1, Ø 4 mm; Henry Schein). Finally, the infiltrated lesions were light-cured for 40 s (Mini LED Curing Light, >1,250 mW/cm$^2$; Satelec Acteon). This infiltration procedure was repeated once (1 min infiltration, 40 s polymerization time).

## External infiltration of Group 2

In line with Group 1, the resin infiltrant for Group 2 (12 premolars/12 molars; n = 24; ICDAS code 2: n = 12, ICDAS code 3: n = 12) was first labeled with a red fluorescent dye (0.1 mmol rhodamine B isothiocyanate, RITC; Babenberger Apotheke); subsequently, the resin was hand-mixed with AgNP (20 nm particle size [48], 5.5 wt%; Ionic Liquids Technologies, Heilbronn, Germany). Infiltration procedure, removal of surpluses, re-infiltration, and polymerization were performed in analogy to Group 1 (see flowchart, Fig 1).

## Preparation of the specimens and microscopic evaluation

Prior to the microscopic examinations, either the buccal or the lingual surface of each tooth was partially ground using a grinder/polisher (MetaServ 250 with Vector Power Head; Buehler, Lake Bluff, IL, USA; CarbiMet, silicon carbide [SiC] grinding paper, P 320; Buehler); this was done parallel to the tooth axis under constant water cooling. Then, the specimens were gently dried with a soft paper towel, and fixed with the grounded surface downside (Sekundenkleber; UHU, Bühl, Germany) on a previously roughened (using SiC abrasive paper (Matador, P 220; Starcke, Melle, Germany)) glass microscopic slide (#190501, 28 × 48 × 1 mm, Menzel-Gläser; Thermo Fisher Scientific, Waltham, MA, USA).

Subsequently, the teeth were cut (IsoMet 1000; Buehler; and IsoMet 15HC metal matrix, #11–4246, 0.5 mm, precision sectioning blade; Buehler) in the mesio-distal direction next to the most carious aspect of the lesion (as previously defined by the laser detection device, see above) under permanent water cooling. Next, each cut tooth was hand-polished (MetaServ 250/Vector Power Head; Buehler; CarbiMet, SiC grinding paper, P 1,200; Buehler; Buehler-Met II, SiC grinding paper, P 2,500; Buehler; MicroCut, SiC grinding paper, P 4,000; Buehler) until the most demineralized area was visible. To ensure that the desired depth range was reached, a light microscope (Nikon SMZ645/Nikon G-AL 1.5×; Nikon, Tokyo, Japan) and magnifying glasses (opt-on TTL 2.7×, 400 mm working distance; orangedental) were used.

After the grinding process, all cut surfaces were cleaned using sodium hypochlorite (NaOCl, 5%; Sigma-Aldrich, Steinheim, Germany; 2 h), to get rid of any smear layer, and to dissolve any organic material potentially occluding the porous lesion volume [57]. Then, the teeth were stored in 50 ml of hydrogen peroxide (30%, Sigma-Aldrich) for another 2 h, to completely dissolve the sodium hypochlorite, and to bleach organic residues remaining in the non-infiltrated enamel pores; this procedure ensured a reduction of the teeth's auto-fluorescence. All teeth were then immersed into tap water at room temperature (4 h). Finally, all surfaces were dried with absolute alcohol (Merck, Darmstadt, Germany) and coated with a blue fluorescent dye (0.1 mmol Berberine; Babenberger Apotheke) using a micro-brush (Microbrush Plus, superfine white, Ø 1 mm; Microbrush International), to allow for a clear differentiation between the infiltrated and the non-infiltrated areas. After 2 h the colorant was rinsed gently into a basin for 1 min, to remove any surface excess of the fluorescent dye.

Using a confocal laser scanning microscope (CLSM; TCS SP8 DMi8; Leica Microsystems, Wetzlar, Germany) and a 10× objective (HC PL Fluotar 10×/0.3 numerical aperture, dry; Leica Microsystems) the lesions of all teeth were captured. The format of the images was individually chosen depending on the size of each lesion. To obtain two-dimensional images, the xy-scan modus was combined with separate extinction settings (561 nm–red/RITC; 520 nm–green/FITC; 405 nm–blue/Berberine). For image analysis, an open source software tool (GIMP 2.8.16 GNU Image Manipulation Program; https://www.gimp.org) was used. By means of GIMP's 'Paths Tool', each lesion was outlined. Then, with each tooth 10 measuring points were randomly selected within the outlined lesion of the red, the green, and the blue image using the 'Color Picker' tool to evaluate the individual fluorescence values for FITC, RITC, and Berberine. Subsequently, the thresholds for RITC, FITC, and Berberine were adjusted excluding the abovementioned tooth-specific individual fluorescence values to use the 'Histogram Dialog' tool aiming to obtain the number of pixels within the outlined area of demineralized enamel and of the infiltrated demineralized enamel areas as well as the percentage values of the infiltration areas.

### Measured variables and statistical analysis

Raw data (recorded in pixels) were entered into Excel sheets (Microsoft, Redmond, WA, USA). Both the total lesion sizes of enamel ($TLS_{Enamel}$) and the infiltrated lesion areas ($ILA_{Enamel}$) were now fractionized with regard to the used fluorescents (FITC, RITC, or Berberine), and the percentages of proportions of infiltrated lesion areas (% $ILA_{Enamel}$) were computed (% $ILA_{Enamel} = ILA_{Enamel} \times 100 \div TLS_{Enamel}$); the same was calculated with the non-infiltrated lesion areas. All statistical analyses were performed by means of a statistical software package (IBM SPSS Statistics 25; IBM Analytics, Armonk, NY, USA), including calculation of means, medians, standard deviations, and quartiles. Homoscedasticity (equality of variances) of the relevant parameters (total lesion size with regard to either tooth type or ICDAS code) was assessed by means of the Levene's test, and the latter was used to test for possible differences in sample variances of the various infiltrated lesion areas ($ILA_{Enamel}$ with regard to FITC, RITC, or Berberine; with or without the use of AgNP). After testing for normal distribution according to the K-S-test (Lilliefors approximation), subgroup comparisons were analyzed using independent *t*-test statistics, and equalities of means were assessed. With all statistical comparisons, significance levels of 5% ($\alpha = 0.05$) indicated significant differences that were unlikely to have arisen by chance.

## Results

During the preparation procedures for histological validation, 5 teeth were lost due to irreparable damage, thus resulting in 21 premolars and 22 molars (ICDAS code 2: n = 21, ICDAS code

3: n = 22) to be processed. According to Levene's test, the homoscedasticity could not be refused (minimum p-value was 0.224), thus revealing comparable total enamel lesion sizes, regardless of tooth types or ICDAS codes. Therefore, the various subgroups could be merged, now considering conventional infiltration (Group 1; n = 19) or infiltration with additional use of AgNP (Group 2; n = 24) as the remaining subgroups.

The internal infiltration procedure (using FITC as fluorescent marker) resulted in substantially infiltrated lesion areas, with a mean of some 50% of the $TLS_{Enamel}$ being infiltrated after bleaching of possibly fluorescent organic remnants. The *t*-test statistics did not reveal any significant differences between Group 1 and Group 2 (p = 0.838; see Table 1). Regarding the external infiltration (using RITC), both groups displayed infiltrated lesion areas with a mean of some 20% of the $TLS_{Enamel}$, again without any significant differences (p = 0.109; Table 1); the outer lesion areas obviously were totally occluded by the resinous infiltrant.

None of the lesions of both groups was completely infiltrated; a representative confocal laser scanning micrograph is provided with Fig 3, depicting the non-infiltrated areas in blue (Berberine). The statistical analysis revealed that both groups did not differ significantly with respect to the percentage distribution of non-infiltrated areas (p = 0.965; Table 1). This could be confirmed by cross-checking with the total infiltration areas (internal plus external infiltration, FITC plus RITC); here, again, no significant differences could be assessed by the statistical analysis either (p = 0.381; Table 1).

## Discussion

With the present study, we introduce a refined treatment concept for proximal lesions clearly extending into dentin. From recent investigations, it is known that dentin subjacent to natural proximal enamel caries is not simply sclerotic (as has been presumed sometimes); instead, these dentin portions have been shown to be demineralized in wide parts [58], thus substantially reducing mechanical resistance, and additionally jeopardizing the integrity of the overlying (and likewise weakened) enamel. Very comparable observations have been reported for occlusal caries lesions reaching to the underlying dentin [59]. Therefore, removal of demineralized dentin (and replacement by adequate restorative materials [60]) should be an appropriate treatment option with these lesions.

This procedure would seem accompanied by a bacterial eradication. No doubt, microbial infection of dentin must be assumed with (micro-)cavitated proximal lesions [34, 61], but the level of infection at the cervical cavity floor of proximal lesions can be greatly reduced, both with conventional and with tunnel cavity preparations [62], thus confirming that removal of soft and infected dentin is effective when pursuing a positive patient outcome. Concerning the conventional infiltration therapy, it must be emphasized that due to the barrier created by an external resin infiltration, resistance to external acids will be increased [5, 23] (even if the infiltrated surface is not completely impervious to a new cariogenic challenge [25, 26]), and outside microorganisms will be expelled. From previous investigations it is known, however, that bacteria can be found within enamel lesions at an early stage of caries development, even with lesions revealing macroscopically intact surfaces [63]. Since bacteria control of deeper intralesional aspects has not been investigated up to now, at best, some positive indirect conclusions by analogy from studies referring to sealing of occlusal caries would seem permissible [64], and this would suggest that it is not necessary to remove all carious dentin prior to placing the restoration; over time, sealing of carious dentin obviously will result in lower (time- and material-specific [65]) levels of infection if compared to the traditional concepts aiming at complete dentin caries removal [66]. Certainly, additional research to elucidate the fate of microorganisms hemmed by the external resin infiltration of proximal lesions is required.

**Table 1. Infiltrated lesion areas (in %) in relation to total enamel Lesions.**

| | $ILA_{Enamel}$ (Internal Infiltration, FITC), % of $TLS_{Enamel}$ | | | | | |
|---|---|---|---|---|---|---|
| | Mean | SD | Median | q1 | q3 | Sig. |
| **Group 1** Infiltration without AgNP | 54.74 | 16.26 | 56.00 | 42.00 | 63.00 | p = 0.838 |
| **Group 2** Infiltration with AgNP | 55.88 | 19.21 | 58.00 | 41.25 | 70.50 | |
| | $ILA_{Enamel}$ (External Infiltration, RITC), % of $TLS_{Enamel}$ | | | | | |
| | Mean | SD | Median | q1 | q3 | Sig. |
| **Group 1** Infiltration without AgNP | 18.84 | 8.30 | 16.00 | 12.00 | 24.00 | p = 0.109 |
| **Group 2** Infiltration with AgNP | 26.29 | 20.10 | 20.50 | 14.25 | 33.50 | |
| | Non-infiltrated Lesion Area (Berberine), % of $TLS_{Enamel}$ | | | | | |
| | Mean | SD | Median | q1 | q3 | Sig. |
| **Group 1** Infiltration without AgNP | 71.95 | 12.20 | 76.00 | 61.00 | 82.00 | p = 0.965 |
| **Group 2** Infiltration with AgNP | 72.13 | 14.08 | 72.00 | 61.25 | 85.75 | |
| | $ILA_{Enamel}$ (Total Infiltration, FITC + RITC), % of $TLS_{Enamel}$ | | | | | |
| | Mean | SD | Median | q1 | q3 | Sig. |
| **Group 1** Infiltration without AgNP | 73.58 | 22.11 | 76.00 | 54.00 | 86.00 | p = 0.381 |
| **Group 2** Infiltration with AgNP | 82.17 | 37.32 | 77.00 | 56.50 | 98.00 | |

$ILA_{Enamel}$, infiltrated lesion area of enamel; $TLS_{Enamel}$, total enamel lesion size of enamel; statistical parameters (means, standard deviation [SD], medians, first [q1] and third [q3] quartiles) as well as exact p values [Sig.] are given for both the internally and the externally infiltrated areas, along with the non-infiltrated lesions areas and the totally infiltrated lesion areas.

Next to the mechanistic and biologic thoughts given above, it should be underlined that the demineralized dentin portion subjacent to a proximal enamel lesion must be expected to provide enlarged pathways for dentinal fluids to penetrate into the proximal caries [58], thus hampering the latter from complete drying, competing with the low-viscosity infiltrant resin, and rendering a complete infiltration impossible from a clinical point of view. This would explain that external resin infiltration alone probably will not be able to completely permeate into the full-thickness of a lesion [7], all the more so as the percentage penetrations of the resin with proximal caries considerably exceeding the enamel-dentin border is notoriously lower than with lesions exclusively restricted to enamel [67]. Thus, even if the technique has been rated effective in arresting the progression of non-cavitated proximal caries involved in the enamel-dentin junction, it would not seem astonishing that for proximal caries clearly exceeding the enamel-dentin border, the therapeutic efficacy of solely external resin infiltration clearly is limited. This has been shown with recent studies focusing on deeper proximal caries [7, 10], thus shedding some ambiguous light on possible survival of trapped microbiota (and their acid production as well as, beyond, their proteolytic and hydrolytic activities). Consequently, this would suggest that due consideration is mandatory with treatment decisions on proximal caries involving the vicinal dentin.

The internationally accepted "gold standard threshold" for minimally invasive operative interventions currently refers to proximal lesions radiographically extending beyond the outer third of dentin (with recommendations aiming at predominantly saucer-shaped preparations as the favored cavity design) [68]. This mainly is owed to the fact of increasing cavitation probabilities of the outer enamel surface [33, 34, 69], and these breakdowns, in turn, correspond to impaired biofilm removal and decreasing remineralization capabilities (even with patients showing normal salivary function [70]); moreover, it is well-known that lesions extending clearly beyond the enamel-dentin border ($> 0.5$ mm) are most likely to progress within a period of 3 years [71], and comparable deteriorations have been reported with increasing

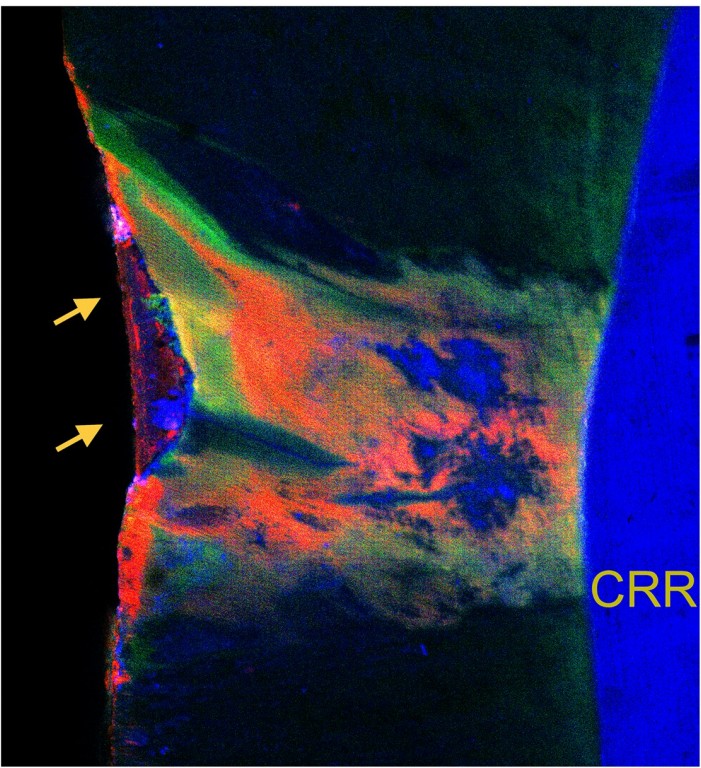

**Fig 3. Merged confocal laser scanning micrograph.** Micrograph (10× magnification) corresponding to the specimen known from Fig 2, and revealing the deep and partially inhomogeneous penetration of the resin infiltrant into the lesion body, visualized by the fluorescently labeled infiltrant resin (internal infiltration with FITC-labeled resin, green; external infiltration with RITC-labeled resin, red; Berberine filling the porous volume, blue), at the same time depicting interdiffusion zones of internal and external infiltration. Note the adhesive seal of the restoration (right), and the partially filled surface damage of the lesion (left part, see arrows). [CRR, composite resin restoration; FITC, fluorescein isothiocyanate; RITC, rhodamine B isothiocyanate.].

ICDAS severities [36]. Therefore, in accordance with the current study set-up, temporary separation of teeth to inspect the proximal surface integrity [72] and DIAGNOcam readings to scrutinize lesion severities [56, 73] would seem advisable prior to any treatment decision.

With proximal caries extending into dentin, modified partial [74] or internal [30] tunnel restoration techniques (both actually representing a Class I cavities), allowing for an internal and external resin infiltration have been suggested recently [32]; it should be emphasized that this concept of double-sided resin infiltration would seem consistent with the prevailing recommendations on treatment decisions related to surgical intervention. Indeed, the tunnel preparation approach (even if challenging with regard to caries removal [74]) dispenses with full surgical intervention as the last resort [14], and aims at a complete preservation of the proximal enamel, including the previously demineralized lesion, now reinforced by means of the double-sided infiltration [32]. Although the non-filled polymerized infiltrant resin itself reveals a low microhardness (even after accelerated aging) [37], it is noteworthy that the reported increase of surface microhardness of infiltrated enamel lesions [17–22, 25] is comparably high; this obviously is due to a uniform complex composed of triethylene glycol dimethacrylate (TEGDMA) and hydroxyapatite, and this interaction with crystals results in improved mechanical strengths [22] and aesthetic appearance [2, 23].

With an internal penetration depth obstructing more than 50% of the $TLS_{Enamel}$ (see Table 1), the current study indeed revealed a considerable portion of internally occluded

carious enamel after a conventional total etch procedure using phosphoric acid, and this clearly outperformed our recently published outcome of internal infiltration [32]. Indeed, internal infiltration obviously should be the first step, and occlusion of the tiny pores might even be enhanced by applying active pressure with the aid of packable composite resins prior to polymerization [75]. The infiltrant resin used in the present investigation (Icon Caries Infiltrant; DMG) has been shown to be compatible to generally utilized adhesive restorations [32, 76], and does not impair the shear bond strengths to dentin [77], while adhesion to sound [78–80] or demineralized [40, 78, 80, 81] enamel even was increased (if compared to other adhesive systems); moreover, tensile bond strength testing was accompanied by a high portion of mixed (cohesive in enamel) failures [80], thus indicating a strong and reliable enamel hybrid layer composed of resin tags enveloping the enamel crystallites [82]. These aspects obviously indicate that final composite resin restorations can be bonded adhesively to the resin-infiltrated hard substances, and this should result in stable repairs, thus complying with the concepts of minimum intervention dentistry [1, 2], and solving (or at least minimizing) some of the clinical problems discussed above.

In the past, tunnel restorations often have been rated with some reservation, and several reasons did account for this conservative or reticent attitude. First, and this is considered important with respect to the current study, the majority of the previous studies investigating the tunnel technique used glass ionomers as restorative materials. With regard to adhesive effects and reinforcement, these must be classified as inferior if compared to composite resins [83]. Consequently, the latter have shown more promising results, with positive laboratory evaluations [83–85], and with high clinical success rates (but with short observation periods of up to 2 years only) [31, 86]. Second, and this should be a consequence of the material-related aspects, former failures were mainly due to fractures of the marginal ridges [87]. Thus, to maintain tooth strength and integrity, an intact marginal ridge should be preserved whenever possible [88]. Therefore, when complying with some preparation guidelines (height of marginal ridge [83–85], width of marginal ridge [83, 89–91], and cavity size [92]), internal tunnels (Class I cavities) should be a feasible treatment option [30]. Additionally, adhesive composite resins exerting strengthening properties like resistance to fracture, failure mode, or stress distribution [93] similar to the unaltered tooth (with a 0.87 relative stiffness [94]) should have a positive effect on fracture resistance, even with reduced marginal ridge dimensions [83, 89]. Finally, further deterioration of the enamel lesion has been blamed to be responsible for the frequent failures of tunnel restorations [95], so both hampering the undermining lesion progress and stabilizing the demineralized lesion by creating an enamel hybrid layer [82] using the internal/external resin infiltration approach should help to maintain tooth rigidity. Nevertheless, while the thoughts given above would seem plausible at a first glance, confirmation of an extensive compatibility of timely adhesive restorative materials to sound dentin (with its functional aspects of elasticity, rigidity, and toughness) would seem mandatory [90], and the clinical efficacy of the approach presented with the current paper has not been corroborated up to now.

In the present study, proximal surfaces revealing initial caries lesions (ICDAS 2) or those considered at the transition to moderate lesions (ICDAS 3) [14, 15] have been used; these lesions can be considered as fast progressing ones, and a therapeutic intervention has been recommended sooner rather than later [36]. Along with the radiologic diagnosis, we used the DIAGNOcam (KaVo) readings for the final tooth selection, since near-infrared transillumination clearly enables differentiation of lesions limited to enamel from those reaching the dentin. It should be kept in mind that bitewing radiographs tend to underestimate lesion depths [55, 96], even if combined with visual inspection [56]; instead, the use of near-infrared

transillumination does disclose the true lesions extent more accurately [73], and, although not considered our primary aim, this has been corroborated by the current study.

Additionally, treatment needs (and in particular the localization of the sectioning area) were assured by means of DIAGNOdent pen (KaVo) with the present set-up. From a histological point of view, enamel lesion sizes proved to be comparable, and, as expected [36], all lesions used in the current investigation had reached the dentin, and turned out to be accessible both for internal and for external infiltration, while this was neither influenced by tooth types nor by ICDAS codes. Thus, clustering of study subsets with respect to the use of AgNP was possible, and the statistical evaluation did not reveal any differences between the both groups, neither regarding the external infiltration nor with reference to the combination of internal and external infiltration. Moreover, all lesions revealed non-infiltrated areas to some extent; concerning this matter, no significant differences could be revealed as well (see Table 1). Within the scope of an overall view, adding AgNP to the infiltrant resin did not affect the infiltration ability of the latter. Consequently, under the limitations of the present *ex vivo* study, the null hypothesis was not rejected.

With the current set-up, we assessed infiltrated lesion areas by evaluating the microscopic images in due consideration of the respective fluorescence modes (and not by analyzing the merged overlays, compare Fig 3). It should be mentioned, that the merged microscopic images showed overlaps of the different fluorescents (FITC, RITC, Berberine). Due to this fact, it can be concluded that the individual pores of the respective lesions having been stained either red (RITC) or green (FITC) have been incompletely infiltrated; presenting some blending with the blue Berberine, this would suggest that the latter was able to penetrate into residual pores previously occluded by organic compounds or trapped air [57], and obviously not completely filled by the polymerized low-viscosity infiltrant resin. However, to the best of our knowledge, we did not find any further information regarding this aspect in the literature, and more research to evaluate the exact distribution of the respective infiltrant portions would seem mandatory.

Recent studies have revealed that organic matter deposited in the porous lesion volume will hamper both remineralization and resinous infiltration of subsurface lesions [57], and removal of this organic debris has been considered pivotal for successful treatment [39]. Therefore, prior to infiltration, the outer surfaces had been deproteinized by means of sodium hypochlorite (5%; 20 min), as has been recommended previously [32, 39, 97]. This (unusually) prolonged cleaning procedure aimed to dissolve organic surface remnants possibly occluding the tiny lesion pores serving as entrance and pathways for the low-viscosity infiltrant resin, and also removed extrinsic deposits possibly covering the micro-cavitations [53], thus appropriately preparing the lesion surface for hydrochloric acid etching [39]. The latter approach will lead to erosive enamel loss [98] and will additionally roughen the surfaces (in particular those with advanced ICDAS codes) [41], whereas the intentionally non-polished infiltrant resin itself obviously is not completely able to smooth out the surface irregularities to an acceptable clinical level (and not comparable to sound enamel) [18–22, 25, 39]; this roughness even should further deteriorate after aging [18]. Representing bi-functional methacrylate monomers, the non-filled diluent TEGDMA component of the infiltrant resin will be inhibited from complete polymerization by free radical scavengers such as oxygen, and loss of this oxygen-inhibited layer might contribute to an increasing roughness with time. Therefore, while finishing procedures will not improve the quality of these rough surfaces [41], bacterial colonization will not be impeded by the infiltrant resin or its main components [17, 19, 42, 99].

While TEGDMA leached from fresh or undercured resin initially (up to 24 h) reduced biofilm metabolic activity (but not biomass) [99], polymerized TEGDMA is prone to water and/ or ethanol sorption leading to monomer hydrolysis and fractures of ester bonds [37], and this

effect is higher than with other widespread resins [100]. Due to internal stress caused by thermal expansion and contraction, these volumetric changes might result in surface microcracks and microfissures (thus providing entrance for fluids and enzymes), as has been recently reported with resin-infiltrated enamel lesions [18]. Salivary [101] and microbial [102] esterase activities (cholesterol esterase, pseudo-cholinesterase) have revealed hydrolysis of TEGDMA, and in particular the matrix biodegradation promoted by bacterial esterases seems to be responsible for the increased composite surface roughness upon biofilm exposures [103]. It should be noted that hydrolytic degradation (by firmly bound water in the enamel pores and water from saliva) yields a hydrophilic product, triethylene glycol (TEG), which has been reported to stimulate growth and pathogenicity of *Strep. mutans* and *Strep. salivarius* [104, 105], thus again leading to increased biofilm accumulation. With the aspects given above, the hydrophilic infiltrant resin is considered a low-level resistant material with progressively reduced mechanical properties [37], judged vulnerable to deterioration in the oral environment [23], and revealing a time-dependent reduction of microhardness [106]. Whereas the otherwise non-controversial and favorable characteristics of TEGDMA as an infiltrant resin should not to be pilloried, the context presented above would seem to elucidate the limited efficacy of resin infiltration with deeper proximal caries lesions [7, 10].

Undoubtedly, strategies including optimized oral hygiene and prudent dietary control are considered paramount to control caries, as has been shown previously with long-existing white spot lesions located on smooth surfaces, which proved to be stable under the conditions of adequate mouth hygiene [107]. Initial proximal lesions, however, are neither clearly visible nor adequately assessable in the majority of cases, and this in particular comes true with progressed caries revealing surface (micro-)breakdown [33, 34], which renders cleansing impossible (compare Fig 2A). Thus, the concept developed for the current study was to combine the infiltrant with an antibacterial agent, such as nanosilver; this additive should safeguard the infiltrated (but still rough) surface areas from *de novo* microbial colonization and subsequent biodegradation.

It should be kept in mind that proximal caries lesions and conventional restorative treatment options will constitute microbial proximal invasions, considered suspicious of negatively affecting outer natural tooth surfaces, and responsible for long-term occlusal and periodontal sequelae [108]. The internal tunnel approach combining internal and external (double-sided) infiltration of the enamel lesions as studied in the current investigation should help to overcome the problems of proximal lesions progressed into dentin and hitherto not designated to the external (single-sided) infiltration approach, thus sustaining the tooth's proximal outline, and delaying the reparative treatment cycle usually associated with conventional Class II preparations (with drawbacks like marginal excess or gaps of filling materials, poor proximal polish of restorations, mal-contouring, complex refurbishment of proximal contact areas, accidental injuries of neighboring teeth, and/or impaired periodontal health [2, 32]).

As with other restorative treatments, the primary aim of infiltration therapy is to facilitate biofilm control. In the present study, external infiltration of the infiltrant resin was not hampered by the nanosilver particles, and reached percentage values of up to some 25% of the $TLS_{Enamel}$. While this was considered comparable to our previous study [32], AgNP concentrated outside the lesion, and enriched in locations of tiny surface disintegration. The latter effect was comparable to recent observations using the same infiltrant resin along with pre-polymerized methacrylate-based nanofillers [109], and clearly revealed a segregation of the components during the infiltration process (see Fig 3). This targeted accumulation is considered advantageous; it should be borne in mind that a reduced biofilm mass would render the contact-killing mechanisms of nanosilver particles effective. Interestingly enough, AgNP have been described to significantly suppress the growth of *Strep. mutans* and *Lactobacillus* with

down to 1% by weight added to commercial composite resins [48, 49], thus confirming the anticariogenic outcome observed with high AgNP concentrations of up to 7 wt% [110, 111].

Hence, in accordance with those previous studies [110, 111], we have chosen 5.5 wt% of AgNP, to vet any possible detrimental effects of the nanoparticles on penetration ability of the infiltrant resin. It is known that, next to the long-term bactericidal activity and the decreased lactic acid production driven by the nanosilver [50, 112, 113], AgNP increased the surface hydrophobicity of composite resins, even with low mass fractions of down to 0.3 wt% [114]. Additionally, it is worth mentioning that with these low concentrations, mechanical characteristics (like flexural strength, or elastic modulus) of the composites matched those of commercial products [112, 113], even though some reports have indicated an influence on polymerization, thus increasing the amount of elutable monomers [115]. This would seem to indicate that the addition of AgNP should not alter the properties of the well-polymerized infiltrant resin. In total, the conceptual shift presented with the current proof-of-concept study should take control over lesion activity of proximal caries lesions extending into dentin (and already revealing minor surface breakdown); along with a monitored preventive regimen, this combined concept of managing the biofilm-mediated and diet-modulated, multifactorial disease called dental caries would seem successful, even over the long term [14, 60]. Though the aspects outlined above have not been evaluated up to now, the underlying background rationale would seem justified to increase dental health, and further studies undoubtedly would seem warranted.

## Conclusion

From the current *ex vivo* investigation, it can be concluded that the ability of the studied infiltrant resin to infiltrate into non- and micro-cavitated proximal enamel caries progressing into dentin will not be negatively impacted by the addition of AgNP. Based on these observations, it would seem reasonable to deduce that the antibacterial effects of AgNP-containing infiltrant resin should hamper the re-formation of microbial biofilms, thus increasing the durability of the infiltrated lesion without compromising its mechanical properties, and impeding the development of recurrent or secondary caries. With the presented internal tunnel approach, the combination of internal and external resin infiltration should result in an increased stabilization of the demineralized enamel and should prevent any sacrifice of sound enamel, thus fostering the transition of dentistry to a minimally invasive and disease-based discipline, and reducing the fatal and avoidable cycle of re-dentistry.

## Supporting information

**S1 File.**
(PDF)

## Acknowledgments

Exceptional thanks go to Monika Altmann-Schnabl (GC Austria, Gratwein-Straßengel, Austria), who made the composite and bonding materials available for the present study. Our gratitude goes to Dipl.-Ing. Dr. techn. Andreas Reisinger (PostDoc, Head of BMLab) and Lukas Warnung, BSc, Lab Technician from the Department of Anatomy and Biomechanics, Division Biomechanics of the Karl Landsteiner—Private University for Health Sciences (Krems, Austria) for their excellent cooperation with the sample preparation. Prof. Dr. rer. soc. oec. Wolfgang Frank (Danube Private University, Krems, Austria) performed the statistical analysis, and this is greatly acknowledged. This investigation was part of a diploma thesis, completed by

Dr. med. dent. Marlene R. Leimer at the Danube Private University. Apart from the support by the authors and their institutions, no external funding was available for this investigator-driven study.

## Author Contributions

**Conceptualization:** Andrej M. Kielbassa.

**Data curation:** Andrej M. Kielbassa, Marlene R. Leimer, Ina B. Ulrich.

**Formal analysis:** Andrej M. Kielbassa.

**Investigation:** Andrej M. Kielbassa, Marlene R. Leimer, Markus Pasztorek, Ina B. Ulrich.

**Methodology:** Andrej M. Kielbassa, Jens Hartmann, Stephan Harm.

**Project administration:** Andrej M. Kielbassa.

**Resources:** Andrej M. Kielbassa.

**Supervision:** Andrej M. Kielbassa, Ina B. Ulrich.

**Visualization:** Andrej M. Kielbassa.

**Writing – original draft:** Andrej M. Kielbassa.

**Writing – review & editing:** Andrej M. Kielbassa, Marlene R. Leimer, Jens Hartmann, Stephan Harm, Markus Pasztorek, Ina B. Ulrich.

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
