## [Decision Letter · Decision Letter 0]

2 Jan 2020

PONE-D-19-27949

Ex vivo  investigation on internal tunnel approach/internal resin infiltration and external nanosilver-modified resin infiltration of proximal caries exceeding into dentin

PLOS ONE

Dear Prof. Dr. med. dent. Dr. h. c. Kielbassa,

Thank you for submitting your manuscript to PLOS ONE. After careful consideration, we feel that it has merit but does not fully meet PLOS ONE’s publication criteria as it currently stands. Therefore, we invite you to submit a revised version of the manuscript that carefully addresses the points raised during the review process.

We would appreciate receiving your revised manuscript by Feb 16 2020 11:59PM. To enhance the reproducibility of your results, we recommend that if applicable you deposit your laboratory protocols in protocols.io, where a protocol can be assigned its own identifier (DOI) such that it can be cited independently in the future. For instructions see: http://journals.plos.org/plosone/s/submission-guidelines#loc-laboratory-protocols

We look forward to receiving your revised manuscript.

Kind regards,

Yogendra Kumar Mishra, Ph. D.

Academic Editor

PLOS ONE

Journal Requirements:

1. 

2. 

We note that you have a patent relating to material pertinent to this article. Please provide an amended statement of Competing Interests to declare this patent (with details including name and number), along with any other relevant declarations relating to employment, consultancy, patents, products in development or modified products etc. Please confirm that this does not alter your adherence to all PLOS ONE policies on sharing data and materials, as detailed online in our guide for authors http://journals.plos.org/plosone/s/competing-interests by including the following statement: "This does not alter our adherence to  PLOS ONE policies on sharing data and materials.” If there are restrictions on sharing of data and/or materials, please state these. Please note that we cannot proceed with consideration of your article until this information has been declared.

Reviewers' comments:

Reviewer's Responses to Questions

**Comments to the Author**

1. Is the manuscript technically sound, and do the data support the conclusions?

Reviewer #1: Yes

Reviewer #2: Yes

2. Has the statistical analysis been performed appropriately and rigorously? 

Reviewer #1: Yes

Reviewer #2: Yes

3. Have the authors made all data underlying the findings in their manuscript fully available?

Reviewer #1: Yes

Reviewer #2: Yes

4. Is the manuscript presented in an intelligible fashion and written in standard English?

Reviewer #1: Yes

Reviewer #2: No

5. Review Comments to the Author

Reviewer #1: The antimicrobial role of nano silver particles in preventing the extension of caries into dentin is very interesting and has potential to control the tooth decay. The work is detailed and technically sound. The results from experiments have been discussed in detail. The manuscript is presented very well. I recommend that the manuscript can be accepted in the current form.

Reviewer #2: 1. The title should be modified to “ Ex vivo investigation on internal tunnel approach/internal and external nanosilver-modified resin infiltration of proximal caries exceeding into dentin”.

2. Line 25-29, Sentence is quite confusing due to grammatical error.

3. Line 28-32, Same confusion.

4. Authors are suggested to rewrite the whole abstract in a concise manner. Try to make short sentences. In order to add more and more information, sentences written are totally not understandable.

5. Introduction is too long and deviating from the study, Moreover same problem is persisting.

6. Disscusion part also is too long although scientifically correct. The author has tried to define each and every aspects with good results. However, the discussion of the results are deviating due to long and length sentences leading to confusion. Moreover, authors are somehow short to justify the purpose of integrating nanosilver.

7. I would recommend authors to go through the manuscript thoroughly and correct English as well as discussion part. The manuscript will than be publishable after a major revision

6. PLOS authors have the option to publish the peer review history of their article (what does this mean?). If published, this will include your full peer review and any attached files.

Reviewer #1: No

Reviewer #2: No

---

## [Author Response · Author response to Decision Letter 0]

10 Jan 2020

Statement of Competing Interests to declare a patent

Andrej M. Kielbassa is appointed as inventor in Austrian, Brazilian, Canadian, Chinese, French, German, Indian, Italian, Japanese, Korean, Russian, Swiss, UK and US patents (held by Charité - Universitätsmedizin Berlin, Germany) for the infiltration technique for cari-ous lesions (“Method and means for infiltrating enamel lesions“, Patent Number: 8853297); these patents have been licensed by DMG (Hamburg, Germany), and Andrej M. Kielbassa receives royalties from this license. This does not alter our adherence to PLOS ONE poli-cies on sharing data and materials.

Reviewers' comments:

Reviewer's Responses to Questions

Comments to the Author  1. Is the manuscript technically sound, and do the data support the conclusions?  The manuscript must describe a technically sound piece of scientific research with data that supports the conclusions. Experiments must have been conducted rigorously, with appropriate controls, replication, and sample sizes. The conclusions must be drawn appropriately based on the data presented. 

Reviewer #1: Yes

Reviewer #2: Yes

Authors: Dear Reviewers, thank you very much for taking some spare time to review our paper.

2. Has the statistical analysis been performed appropriately and rigorously? 

Reviewer #1: Yes

Reviewer #2: Yes

Authors: Thank you.

3. Have the authors made all data underlying the findings in their manuscript fully available?  The PLOS Data policy requires authors to make all data underlying the findings described in their manuscript fully available without restriction, with rare exception (please refer to the Data Availability Statement in the manuscript PDF file). The data should be provided as part of the manuscript or its supporting information, or deposited to a public repository. For example, in addition to summary statistics, the data points behind means, medians and variance measures should be available. If there are restrictions on publicly sharing data—e.g. participant privacy or use of data from a third party—those must be specified.

Reviewer #1: Yes

Reviewer #2: Yes

Authors: Thank you.

4. Is the manuscript presented in an intelligible fashion and written in standard English?  PLOS ONE does not copyedit accepted manuscripts, so the language in submitted articles must be clear, correct, and unambiguous. Any typographical or grammatical errors should be corrected at revision, so please note any specific errors here.

Reviewer #1: Yes

Reviewer #2: No

Authors: Thank you. We have thoroughly revised our manuscript, and we hope that the second Reviewer’s concerns will be satisfied.

5. Review Comments to the Author  

Reviewer #1: 

The antimicrobial role of nano silver particles in preventing the extension of caries into dentin is very interesting and has potential to control the tooth decay. The work is detailed and technically sound. The results from experiments have been discussed in detail. The manuscript is presented very well. I recommend that the manuscript can be accepted in the current form.

Authors: Thank you very much. Your comments are appreciated.

Reviewer #2: 

1. The title should be modified to “ Ex vivo investigation on internal tunnel approach/internal and external nanosilver-modified resin infiltration of proximal caries exceeding into dentin”.

Authors: We have tried to shorten the title in the way this reviewer has suggested. However, this might be misleading. Indeed, internal resin infiltration was done with non-modified infiltrant, while only with the external approach a nanosilver-modified infiltrant was used. We strongly feel that our first version should be more comprehensible for the readers. With 170 characters, the limit set by PLoS ONE (250) will not be exceeded. We do hope that you agree.

2. Line 25-29, Sentence is quite confusing due to grammatical error.

Authors: We have revised the whole text (including the Abstract section), and grammatical shortcomings should have been eliminated.

3. Line 28-32, Same confusion.

Authors: We have revised the whole text (including the Abstract section), and all language shortcomings should be smoothened now. 

4. Authors are suggested to rewrite the whole abstract in a concise manner. Try to make short sentences. In order to add more and more information, sentences written are totally not understandable.

Authors: We have revised the Abstract section, and readability should be improved now (299 words, and not exceeding PLoS ONE’s limits). Thank you for alerting. We hope that this revised version will be satisfying.

5. Introduction is too long and deviating from the study, Moreover same problem is persisting.

Authors: Please note that PLoS ONE does not restrict on word count, and manuscripts can be any length (see https://journals.plos.org/plosone/s/submission-guidelines#loc-references). Notwithstanding, we have thoroughly revised our manuscript, at the same time following the PLoS ONE instructions regarding the Introduction section. Last but not least, with our manuscript a new treatment rationale is presented, and we feel that the respective aspects should be clearly elucidated, to foster a deep understanding.

6. Disscusion part also is too long although scientifically correct. The author has tried to define each and every aspects with good results. However, the discussion of the results are deviating due to long and length sentences leading to confusion. Moreover, authors are somehow short to justify the purpose of integrating nanosilver.

Authors: As with the Introduction section, we have thoroughly revised the Discussion part. Regarding the nanosilver integration, there are several paragraphs elaborating the microbiological challenges and revealing the effects of nanosilver itself. Thus, we feel that the reader will be able to understand the background rationales.

7. I would recommend authors to go through the manuscript thoroughly and correct English as well as discussion part. The manuscript will than be publishable after a major revision

Authors: Please note that we have thoroughly revised the whole manuscript, to smooth out any language shortcomings and to facilitate readability. Once again, thank you very much. Your comments are greatly appreciated.

6. PLOS authors have the option to publish the peer review history of their article (what does this mean?). If published, this will include your full peer review and any attached files.   Do you want your identity to be public for this peer review? For information about this choice, including consent withdrawal, please see our Privacy Policy.

Reviewer #1: No

Reviewer #2: No

Authors: We have already used PACE with our first submission, so this should be OK.

Once again, thank you for your comments. We hope that our revised version will be ready to proceed.

Sincerely,

Andrej M. Kielbassa (on behalf of all co-authors)

---

## [Editor Report · Decision Letter 1]

13 Jan 2020

Ex vivo investigation on internal tunnel approach/internal resin infiltration and external nanosilver-modified resin infiltration of proximal caries exceeding into dentin

PONE-D-19-27949R1

Dear Dr. Kielbassa,

We are pleased to inform you that your manuscript has been judged scientifically suitable for publication and will be formally accepted for publication once it complies with all outstanding technical requirements.

With kind regards,

Yogendra Kumar Mishra, Ph. D.

Academic Editor

PLOS ONE
---

## [Editor Report · Acceptance letter]

14 Jan 2020

PONE-D-19-27949R1 

*Ex vivo* investigation on internal tunnel approach/internal resin infiltration and external nanosilver-modified resin infiltration of proximal caries exceeding into dentin 

Dear Dr. Kielbassa:

I am pleased to inform you that your manuscript has been deemed suitable for publication in PLOS ONE. Congratulations! Your manuscript is now with our production department. 

With kind regards,

on behalf of

Dr. Yogendra Kumar Mishra 

Academic Editor

PLOS ONE